# Patterns of Thyroid Hormone Prescription in Patients with Bipolar or Schizoaffective Disorder: Findings from the LiSIE Retrospective Cohort Study

**DOI:** 10.3390/jcm10215062

**Published:** 2021-10-29

**Authors:** Ingrid Lieber, Michael Ott, Louise Öhlund, Robert Lundqvist, Mats Eliasson, Mikael Sandlund, Ursula Werneke

**Affiliations:** 1Department of Clinical Sciences, Psychiatry, Sunderby Research Unit, Umeå University, 90185 Umeå, Sweden; louise.ohlund@umu.se (L.Ö.); ursula.werneke@umu.se (U.W.); 2Department of Public Health and Clinical Medicine, Umeå University, 90185 Umeå, Sweden; michael.ott@umu.se; 3Department of Public Health and Clinical Medicine, Sunderby Research Unit, Umeå University, 90185 Umeå, Sweden; robert.lundqvist@norrbotten.se (R.L.); mats.eliasson@norrbotten.se (M.E.); 4Division of Psychiatry, Department of Clinical Sciences, Umeå University, 90185 Umeå, Sweden; mikael.sandlund@umu.se

**Keywords:** bipolar disorder, schizoaffective disorder, lithium, mood stabilizer, hypothyroidism, thyroid dysfunction, TSH

## Abstract

The prescription of thyroid hormone replacement therapy (THRT) has increased in the general population; the thyroid stimulating hormone (TSH) threshold to initiate THRT has decreased. It remains unclear whether a similar trend has occurred in patients with bipolar disorder (BD). In this work we explore patterns and trends of prescribing THRT in patients with BD or schizoaffective disorder (SZD) with an observational study and time-trend analysis in the framework of the LiSIE (Lithium—Study into Effects and Side Effects) retrospective cohort study. In most patients, THRT was initiated for subclinical hypothyroidism. The median TSH at which THRT was started was 6.0 (IQR 4.0) mIU/L and the median free serum thyroxine (fT4) at which THRT was started was 11.8 (IQR 3.9) pmol/L. The median TSH concentration at the start of THRT decreased annually with 0.10 mIU/L (*p* = 0.047) and was higher in patients treated with lithium than in patients treated with other mood stabilisers (*p* = 0.02). In conclusion, THRT was typically initiated in the context of mild or absent alterations of thyroid function tests with a decreasing TSH threshold. As THRT is rarely reversed once initiated, clinicians need to weigh up potential benefits and risks when prescribing THRT for subclinical hypothyroidism in patients with BD or SZD.

## 1. Introduction

Thyroid function disorders are common in patients with bipolar disorder (BD). Hypothyroidism is much more common than hyperthyroidism and may be associated with mood-stabiliser (MS) treatment [1,2,3]. Lithium accrues the highest risk, but other MSs, such as some second-generation antipsychotics (SGAs) and anticonvulsants, have also been implicated. Of these, quetiapine seems to carry the highest risk of hypothyroidism, nearly on par with lithium [4]. The condition of BD by itself may also be associated with hypothyroidism [5]. However, the relation between BD and hypothyroidism remains poorly understood [6]. Diagnosing hypothyroidism in patients with BD can be complex. Both conditions are chronic. Furthermore, symptoms of hypothyroidism, such as impaired concentration, lethargy, and psychomotor retardation, can overlap with symptoms of BD [7,8]. Still, thyroid status itself does not reliably account for mood changes in BD patients [9,10,11]. However, hypothyroidism may predict a lesser response to MSs [7,8,12,13,14,15]. At the same time, thyroid dysfunction may increase the risk of manic episodes [16]. Thyroid hormone replacement therapy (THRT) as an augmentation treatment in euthyroid BD patients may improve mood [17] and prolong the time in euthymic state [18]. Yet, in patients with subclinical hypothyroidism, even if thyroid dysfunction affected mood, THRT might still not effectively treat depressive symptoms [19]. Ultimately, the evidence remains conflicting [19,20,21]. In the two most recent decades, the prescription of THRT has increased in the general population in several countries [22,23]. At the same time, the thyroid stimulating hormone (TSH) threshold at which THRT is started has decreased [24]. There is a concern that the prevalence of THRT in patients with subclinical hypothyroidism is at odds with the evidence [25,26]. This could then amount to the over prescription of THRT [23]. Yet, some recent data from the US suggest that the trend toward THRT over prescribing has started to reverse or remains unchanged [26,27].

It currently remains unclear how THRT is used in patients with BD. Equally, we do not know whether a comparable trend of THRT prescription has occurred. Clinicians may prescribe THRT to patients with BD for several reasons, such as: (a) hypothyroidism associated with MS, particularly lithium; (b) explicit augmentation therapy for depression; and (c) an unspecific attempt to influence mood favourably. However, there are caveats. As evidence from the general population suggests, THRT prescribing for subclinical hypothyroidism may neither improve depressive symptoms nor quality of life [28]. The potential risks of THRT may outweigh potential benefits [25]. Yet, once THRT is started, its prescription tends to be long term [24]. The indication for THRT is rarely revisited. This may hold true, even if potentially offending agents such as lithium are withdrawn [29].

### Aims

In view of the current controversies regarding THRT in individuals with subclinical hypothyroidism, we sought to explore patterns of THRT use in patients with BD or schizoaffective disorder (SZD). Specifically, we tested the following three hypotheses:

In the majority of patients with BD/SZD, THRT is prescribed only for mild or no alterations of thyroid function tests (TFT) and/or unspecific symptoms.

The TSH concentration, at which THRT is initiated (TSH_THRT_), has decreased over time.

In patients treated with lithium, TSH_THRT_ is lower compared to other MSs.

## 2. Materials and Methods

### 2.1. Study Design

This study is a part of the LiSIE (Lithium—Study into Effects and Side Effects) research programme, a retrospective cohort study based on a review of medical records. LiSIE aims at identifying the best long-term treatment options for patients with BD and related conditions by exploring the effects and potential adverse effects of lithium compared to other MS. The study was conducted according to the guidelines of the Declaration of Helsinki and approved by the Regional Ethics Review Board at Umeå University, Sweden (DNR 2010-227-31M, DNR 2011-228-32M, DNR 2014-10-32M, DNR 2018-76-32M).

Within the framework of this retrospective cohort study, we use different designs for each hypothesis. For hypothesis 1, we explored thyroid status at THRT initiation. For hypothesis 2, we used a time-trend analysis. For hypothesis 3, we compared patients treated with lithium and patients treated with other MS as case controls.

### 2.2. Lithium—Study into Effects and Side Effects Participants

LiSIE invited all individuals in the Swedish regions of Västerbotten and Norrbotten ≥18 years of age, who had either received, according to the Tenth Revision of the International Classification of Diseases (ICD-10), a diagnosis of BD (ICD F31) or SZD (ICD F25) between 1997 and 2011, or who had used lithium as MS between 1997 and 2011 [29]. We excluded patients who, after manual validation from the medical records, more likely had a diagnosis of schizophrenia than BD or SZD [30]. Participants were informed about the nature of the study in writing and provided verbal informed consent. The consent was documented in our research files, dated, and signed by the research worker who obtained the consent. In accordance with the ethics approval granted, deceased patients were also included. Consent procedures concluded by the end of 2012. The cohort was locked at this point; no new patients were included in the study thereafter.

### 2.3. Patient Selection and Inclusion Criteria

For the current study, we included patients from the region of Norrbotten who had received (a) a diagnosis of either BD or SZD on at least two occasions, at least six months apart any time between 1997 and 2013, (b) at least one THRT prescription (levothyroxine or liothyronine) between 1997 and 2017, and (c) their first THRT prescription after the diagnosis of BD/SZD, or after start of the MS treatment. For hypothesis 1, exploring TFT and thyroid-related symptoms at THRT initiation, we included the whole sample. For hypothesis 2, exploring TSH_THRT_ time trends, we used the sample for hypothesis 1, except patients who (a) had received THRT as augmentation therapy, or (b) had started THRT in relation to pregnancy. For hypothesis 3, examining TSH_THRT_ by MS, we used patients sampled for hypothesis 2, who had received MS treatment consecutively for at least 14 days during the three months before THRT initiation, (Figure 1).

### 2.4. Exclusion Criteria

For the current study, we excluded patients who had received (a) THRT after goitre surgery, (b) an explicit diagnosis of hyperthyroidism/thyrotoxicosis before THRT initiation, or (c) THRT without recording of TFT or information regarding the reasons for THRT initiation [29]. We also excluded patients who, at THRT initiation, were less than 18 years old, (Figure 1).

### 2.5. Outcome Definition

#### 2.5.1. Thyroid Status at Which Thyroid Hormone Replacement Therapy Was Started

The main outcome for all hypotheses was thyroid status at which THRT was started. For patients who had not received THRT for augmentation, we categorised thyroid status into five categories: normal, overt hypothyroidism, subclinical hypothyroidism, low free serum thyroxine (fT4), unclassified/other. Subclinical hypothyroidism was stratified into grade 1 and grade 2 depending on TSH levels [31] (Table 1). We classified thyroid status categories according to the laboratory methods and reference intervals used at the time. Laboratory methods were known for 91.6% of tests. Reference intervals were known for all tests, allowing accurate categorisation of thyroid status (Appendix A). Most laboratory values were analysed with an immunoassay from Roche Diagnostics Scandinavia with normal range reference values for thyroid function tests of 0.27–4.20 mIU/L for TSH and 12.0–22.0 pmol/L for fT4. In case of repetitive TFT, we assessed TFT closest before the initiation of THRT.

#### 2.5.2. Reasons for Thyroid Hormone Replacement Therapy Initiation

For hypothesis 1, we checked the medical records for any documented reasons for THRT initiation. Reasons included (a) laboratory findings indicating thyroid dysfunction, (b) somatic symptoms attributed to hypothyroidism, (c) psychiatric symptoms possibly related to hypothyroidism, (d) explicit psychiatric augmentation therapy, and (e) pregnancy related THRT initiation. In many cases, there was more than one reason. Therefore, we formulated decision drivers resulting from the combinations of the above reasons. As somatic symptoms of hypothyroidism, we recorded fatigue, weight gain, cold intolerance, goitre, hair loss, tremor, constipation and other symptoms: swollenness, eye protrusion, dry skin, autonomic dysfunction. The combination of several somatic symptoms was common. Hence, the number of symptoms exceeded the number of patients. Psychiatric symptoms were deemed possibly related to hypothyroidism if noted in medical records at THRT initiation. As such, we recorded depression/low mood, and other symptoms, including anxiety, irritability and difficulties concentrating. We first established reasons for the whole sample. Then, we stratified further according to decision drivers.

#### 2.5.3. TSH at Thyroid Hormone Replacement Therapy Initiation over Time

For hypothesis two, we established the trend of TSH_THRT_ concentrations between 1997 and 2017. We determined the median TSH_THRT_ for every year sampled. To account for changes in reference ranges during the 20-year review period we did not only rely on TSH and fT4 values in absolute but also in relative terms. For TSH, we calculated the percental deviation from the upper normal range. For fT4, we calculated the percental deviation from the lower normal range.

#### 2.5.4. Time from Starting Mood Stabilisers to Starting Thyroid Hormone Replacement Therapy

For hypothesis three, we determined the time from starting a MS to starting THRT. We traced back the time point from which THRT was started to the first day of the respective episode with continuous MS treatment as defined in Section 2.6. To capture the complete length of MS exposure we also recorded MS episodes back until 1970.

### 2.6. Other Variable Definitions

#### 2.6.1. Subtypes of Bipolar Disorder

LiSIE recruited patients who had a diagnosis of BD (F31) or SZD (F25) according to ICD-10. We included both BD and SZD because disturbances in thyroid function tend to be similar in both conditions. Besides, affective and psychotic symptoms can occur on a continuum [32,33,34,35,36]. We then validated the respective diagnoses further from the medical case records according to the DSM-5 subcategories as used in a previous study [30]. These subcategories included BD-I (296.4), SZD (295.7), BD-II (296.80), and other BD (296.89) according to DSM-5. Absence of manic episodes allocated patients to the BD-II or BD other group by default. As diagnosis could vary over time within these diagnostic categories, we recorded diagnosis nearest to the time of THRT initiation. Thereafter, these were stratified into two groups according to the underlying affective disorder. BD-I and SZD were allocated in one group (BD-I/SZD) and BD-II, BD, or otherwise specified BD (BD-II/BD other), in another group.

#### 2.6.2. Age

We recorded the age at THRT initiation. Based on a previous study on the long-term effects of lithium on thyroid and other organ functions, we stratified age into two categories, <60 or ≥60 years [29].

#### 2.6.3. Thyroid Hormone Replacement Therapy Initiating Clinic

We recorded the clinic that had taken the decision to initiate THRT in three categories: (a) psychiatric clinic; (b) general practitioner (GP); and (c) other.

#### 2.6.4. Mood Stabilisers

In accordance with Lambert et al. [4], we included as MSs: (a) lithium; (b) anticonvulsants, i.e., carbamazepine, lamotrigine and valproate; and (c) the SGAs, such as aripiprazole, olanzapine, quetiapine, and risperidone. To be considered MS-associated, there had to be a temporal relationship between MS exposure and THRT initiation. For this, two conditions had to be met: (a) consecutive MS treatment of at least 14 days duration; and (b) the MS was not discontinued earlier than three months before THRT initiation. We assumed that an association between thyroid dysfunction and MS use was unlikely if MS therapy had been discontinued more than three months before THRT was started. We based this assumption on available experience with amiodarone [37]. We stratified MS exposure into two categories: (a) patients exposed to lithium; and (b) patients exposed to a MS other than lithium. To count as continuous MS use, episodes of MSs had to be within three months of each other.

#### 2.6.5. Mood Stabiliser Combination Therapy at Thyroid Hormone Replacement Therapy Initiation

If patients had only one MS within the three months preceding THRT initiation, we recorded monotherapy. If patients had more than one MS during the three months, we recorded combination therapy. We considered combination therapy a proxy for a more complex underlying affective disorder.

#### 2.6.6. Mood Stabiliser Treatment Stability

We considered MS treatment stable when there was no more than one change of MS regimen in the year before THRT initiation. We considered MS treatment unstable when there were two or more changes of MS regimen that year; this, we regarded as another proxy for a more complex underlying affective disorder.

#### 2.6.7. Other Psychotropic Medications

We also checked for other psychotropic medication that could affect thyroid function, such as tricyclic antidepressants (TCA) and phenothiazines [38,39].

### 2.7. Validation of Data

The dates of the electronic prescriptions when a MS had been started or discontinued were manually validated in the medical records for all patients. This way, we confirmed whether an MS had been used before THRT initiation, and if so, for how long. In the same way, we confirmed the accuracy of the start date and reasons for THRT initiation in the medical records.

### 2.8. Chart Review and Analysis

The medical records of all eligible patients were retrospectively reviewed for the outcomes and variables under study, from 1 January 1997 up until 31 December 2017.

### 2.9. Control for Bias

We checked for selection bias in the whole retrospective cohort study (LiSIE). Age, sex, and where applicable, maximum-recorded lithium and creatinine concentrations were key parameters, available in form. In accordance with the ethics approval granted, we compared these parameters for consenting and non-consenting patients. No significant difference was found between the two groups.

### 2.10. Missing Data

In terms of the reasons for THRT initiation, we extracted all available information for the whole sample. When reasons were not documented, we specified this in the results (Figure 2). For two patients, case records were incomplete, and THRT prescription could not be validated.

### 2.11. Statistics

All data were anonymised before statistical analysis. We first analysed the data descriptively, with means and medians for continuous variables and frequencies for categorical variables. For the analysis of relations between categorical variables, chi-square tests were used. Two-sided Fisher’s exact test was used if a table cell had an expected count of <5. We used the z-test for two proportions to compare the sex distribution of the whole LiSIE cohort with the sex distribution for our sample. For each year under study, we performed time trend analyses for median TSH with quantile regression. We used the Mann–Whitney U test to determine any potential differences in median TSH_THRT_ concentrations and median fT4_THRT_ concentrations according to (a) diagnosis, (b) age category, (c) sex, and (d) MS category. We used Kaplan– Meier plots to map the time from starting a MS to starting THRT. The difference in these curves according to (a) lithium exposure, (b) combination therapy, and (c) treatment stability, was analysed with a log-rank test. Throughout, the significance was set to *p* = 0.05. For the statistical analysis, we used SPSS version 26.0 and 27.0 (IBM, Armonk, NY, USA). We have summarised our method in a STROBE checklist (Appendix A).

## 3. Results

### 3.1. Baseline Characteristics at Time of Thyroid Hormone Replacement Therapy Start

Of 1564 included patients with BD or SZD, 421 (26.9%) had received THRT at some point. Of these, 359 patients started THRT within our review period. A total of 291 patients met our inclusion criteria (Figure 1). For hypothesis one, we included all 291 patients. For hypothesis two, we included a subset of 281 patients. For hypothesis three, we included a subset of 260 patients (Table 2).

### 3.2. ***Hypothesis 1 (H1).*** In the Majority of Patients with Bipolar Disorder/Schizoaffective Disorder, Thyroid Hormone Replacement Therapy Is Prescribed for Only Mild or No Alterations of Thyroid Function Test and/or Unspecific Symptoms

In the 291 patients included, THRT was more common in women (71.8%) than in men (28.2%) as compared with the underlying sex distribution of the LiSIE cohort of 62.1% women and 37.9% men (*p* < 0.001), (Figure 1). The most common thyroid status at THRT start was subclinical hypothyroidism with 40.5% (Table 3). This was significantly more common in men than in women: 52.4% vs. 35.9% (*p* < 0.010). For overt hypothyroidism or unclassified/other types of thyroid function as specified in Table 1, there were no sex differences.

Subclinical hypothyroidism grade 1 was more common than subclinical hypothyroidism grade 2 (*p* < 0.001). Approximately one third of patients had either overt hypothyroidism (26.1%) or a TSH_THRT_ ≥ 10 mIU/L, amounting to grade 2 subclinical hypothyroidism (6.5%). There were seven patients who had received THRT explicitly for augmentation, irrespective of thyroid status. There were 37 patients (12.7%) with normal TFT at the time of THRT initiation. Unclassified thyroid dysfunction at THRT initiation was more common in BD-I/SZD than in BD-II/BD other (*p* = 0.014) (Table 3). There were no significant differences in thyroid function according to age ≥ 60 years and <60 years.

#### Decision Drivers for Thyroid Hormone Replacement Therapy Initiation

TFT results were the most common reason for THRT initiation. In 34.0% of the 291 patients, TFT results were the sole driver for starting THRT. In 37.5% of patients, THRT was started in consideration of TFT results in combination with somatic reasons, psychiatric reasons, or both. In 2.1% of patients, THRT was commenced on the grounds of somatic symptoms alone. A figure of 1.0% received THRT during pregnancy and 2.4% of patients received THRT for augmentation. For 21.3% of the sample decision drivers were not recorded in the medical records. Type of bipolar disorder or MS treatment, lithium or other, did not significantly change the distribution of decision drivers. Aside from augmentation, there were no cases with psychiatric reasons as the only decision driver (Figure 2). However, 11.7% of patients received THRT for psychiatric reasons in combination either TFT and/or somatic symptoms. Depression or low mood was most common with 9.6%. Somatic reasons were recorded at THRT initiation in 36.6% of patients. Among the somatic reasons, fatigue and weight gain were most common (Figure 3). There was no difference in median TSH and T4 between patients for whom only TFT was recorded and for whom TFT and symptoms were recorded.

### 3.3. ***Hypothesis 2 (H2).*** The TSH Concentration, at Which Thyroid Hormone Replacement Therapy Is Initiated, Has Decreased over Time

Here, we excluded patients in whom THRT was initiated for the purpose of augmentation or during pregnancy. This yielded a sample of 281 patients to be considered for our trend analysis. However, the TSH_THRT_ test was only available for 277 patients, and fT4_THRT_ was only available for 251 patients. For patients with available TSH_THRT_, the median TSH was 6.0 (min 0.1, max 74.7, IQR 4.0) mIU/L (Figure 4). There were no significant differences between men and women (*p* = 0.203), subtypes of bipolar disorder (*p* = 0.635), or patients aged ≥60 years or <60 years (*p* = 0.719). For patients with available fT4, the median fT4 was 11.8 (min 2.4, max 24.8, IQR 3.9) pmol/L (Figure 5). Men had significantly higher median fT4 than women (12.8: 11.6 pmol/L, *p* = 0.005). There were no significant differences between subtypes of bipolar disorder (*p* = 0.521), or patients aged ≥60 compared to <60 years (*p* = 0.709).

During our 20-year review period from 1997 to 2017, median TSH_THRT_ decreased significantly over time (Figure 6). Median TSH decreased yearly with 0.10 mIU/L (*p* = 0.047). In almost all cases, 95.4%, THRT was initiated by psychiatric clinics or GPs. There was no significant difference regarding the median TSH_THRT_ between GPs and psychiatric clinics (5.5 vs. 6.1 mIU/L *p* = 0.909).

### 3.4. ***Hypothesis 3 (H3).***In Patients Treated with Lithium, the TSH Concentration, at Which Thyroid Hormone Replacement Therapy Is Initiated, Is Lower

Of the 260 patients with MS included in this analysis, 82.3% were exposed to lithium as either a mono- or combination therapy during the three months before THRT initiation (Table 2). In the group exposed to lithium, the TSH_THRT_ median was 6.1 (min 0.1 max 72.0, IQR 4.98) mIU/L. In the group not exposed to lithium, the median TSH_THRT_ was 4.5 (min 0.9, max 22.9, IQR 3.71) mIU/L. Median TSH_THRT_ was significantly higher in patients exposed to lithium (*p* = 0.02). The median time from MS start to THRT initiation was 2.4 years (min 14 days, max 29 years, IQR 4.39 years), (Figure 7). There was no significant difference between lithium and other MSs regarding time between MS start and THRT initiation (*p* = 0.747), (Figure 8).

To explore hypothesis three further, we also examined the pattern of THRT initiation in terms of complexity of the underlying mood disorder, regarding (a) MS monotherapy versus MS combination therapy, and (b) MS treatment stability.

#### Mood Stabiliser Therapy

There were 120 (46.2%) patients with MS monotherapy. Of these, 78.3% had lithium monotherapy. 140 (53.8%) patients had MS combination therapy. Lithium and olanzapine were the most common combination encountered. Median TSH_THRT_ was not significantly higher in patients with MS monotherapy (*p* = 0.527). Neither was there any statistically significant difference in median fT4_THRT_ between both groups (*p* = 0.830). There were 185 (71.2%) patients with stable MS treatment, defined as no or only one change of MS in the year before THRT was initiated. There were 75 (28.8%) patients with unstable MS treatment, defined as two or more MS changes in the year before start of THRT. There were no statistically significant differences in median TSH_THRT_ or median fT4_THRT_ between patients with stable and unstable treatment (*p* = 0.837, *p* = 0.303).

## 4. Discussion

To our knowledge, this is the first study exploring thyroid status at THRT initiation in patients with BD or SZD. We found that subclinical hypothyroidism was the most common thyroid status at THRT initiation. Our findings therefore confirm hypothesis one. In our study, THRT overall was more common in women. However, subclinical hypothyroidism at THRT start was more common in men. Thyroid status may also be age dependent. In one study of patients >65 years treated with lithium for BD or as an adjunct for depression, subclinical hypothyroidism was more common in women [40]. Ultimately, findings from the literature remain heterogeneous with studies using varying TSH cut-off points for hypothyroidism [9,41,42], and hence difficult to interpret.

Concerning thyroid status, we specifically explored the relevance of the type of affective disorder: BD-1/SZD compared to BD-II/BD other. Only non-specific alterations of thyroid status were more common in patients with BD-1/SZD. The clinical significance remains unclear; we found no other differences in thyroid status between the two patient groups. Hypothyroidism and altered thyroid status are common in patients with BD/SZD. For patients with BD/SZD, studies published since 2010 report prevalence rates of 4–32% for subclinical hypothyroidism and of 4–24% for overt hypothyroidism. Lithium exposure contributed to higher hypothyroidism prevalence in these studies [32,33,40]. Regarding the general population, one Swedish study based on patient interviews about the life-time occurrence of hypothyroidism found a prevalence of 4.4% [43]. A Norwegian population-based study reported a prevalence of hypothyroidism of about 6.4% during 2006–2008. Hypothyroidism was defined as a single measurement of TSH above 4.5 mU/L or being treated with THRT [44]. Subclinical hypothyroidism with TSH elevations may, however, be transient. TFT can normalise without THRT. A prospective study followed the thyroid status of 3996 US individuals, ≥65 years of age, over four years. At the starting point of the study, there were 3594 individuals who were not treated with THRT at the time. Of these, 459 (12.8%) had subclinical hypothyroidism, defined as a TSH of 4.5–19.9 mU/L with normal fT4. At the two-year follow-up, of 369 individuals still available, 128 (35%) had reverted to a euthyroid state; 208 (56%) remained subclinically hypothyroid. Of the 128 individuals who had reverted to euthyroidism at year two, 48% remained euthyroid at year four and 32% were again subclinically hypothyroid. Of the 208 individuals who had remained subclinically hypothyroid at year two, 8% reverted to a euthyroid state at year four [45].

In our study, TFT results were the most common driver for THRT initiation. At present, it remains unclear to what extent minor disturbances of thyroid dysfunction can adversely affect mood [19,20,21]. Two meta-analyses have explored the effectiveness of THRT in individuals with subclinical hypothyroidism. The first meta-analysis found no clinically relevant benefits of THRT on depressive symptoms, fatigue, or overall quality of life [28]. In the light of this, an international expert panel concluded that adults with subclinical hypothyroidism generally would not benefit from treatment with thyroid hormones [25]. The second meta-analysis examined a potential relationship between subclinical hypothyroidism and depression in three ways. Individuals with subclinical hypothyroidism had a higher risk of depression than their euthyroid counterparts. There was no difference in TSH in individuals with or without depression. THRT did not lead to significant improvement of depression scores [19]. Therefore, even if subclinical hypothyroidism affects mood, underlying depressive symptoms may not be treatable with THRT. Subclinical hypothyroidism is, however, associated with heart disease, such as coronary heart disease or heart failure [46,47,48,49]. One randomised controlled trial (RCT) of 737 individuals with persisting subclinical hypothyroidism and ≥65 years of age did not find any impact of THRT on fatal or nonfatal cardiovascular events at the one-year follow up [50]. Ultimately, treating subclinical hypothyroidism becomes a matter of balancing potential risks and benefits.

Few studies have explored the use of THRT in patients with BD/SZD. Results are mixed. Previous studies have suggested that thyroid status may affect treatment response to mood stabilisers [7,15]. It has also been suggested that alterations in thyroid status may increase the risk of mania [16] and impair cognitive function [51]. One RCT compared supplementation with triiodothyronine (T3) or thyroxine (T4) against a placebo in 32 patients with treatment-resistant rapid cycling BD. T4 supplementation led to a longer time in a euthymic state and less time in a mixed state. There were no differences between patients receiving T3 or placebo [18]. In a retrospective chart review in 159 patients with BD-II/BD other, 84% of the patients experienced improvement of T3 augmentation [17]. A recent RCT explored adjunctive treatment with supraphysiologic T4 doses in patients with bipolar depression. Compared to placebo, there was no statistically significant difference in reported anxiety, the primary outcome of this study [52]. Other observational studies have also failed to show any association between laboratory thyroid status and mood states in patients with BD [9,10,11].

Regarding hypothesis two, we found that median TSH_THRT_ significantly decreased by 0.1 mIU/L per year during the 21-year, follow-up period. Our findings therefore confirmed hypothesis two. Similar results have been reported in the general population. A Danish register study found that median TSH_THRT_ decreased from 10 mIU/L to 6.8 mIU/L over a 15-year period, which would amount to an annual decrease of 0.21 mIU/L of median TSH_THRT_. In 2015, the final year of the study, 25% of patients had TSH_THRT_ <5 mIU/L [53]. A UK register study found a similar, but flatter, trend. In this study, median TSH_THRT_ decreased from 8.7 mIU/L to 7.9 mIU/L [24]. This would correspond to an annual decrease of 0.1 mIU/L of median TSH_THRT_, precisely in accordance with our study. Another population-based study from the US, however, found a non-significant decrease in median TSH_THRT_ from 5.8 mIU/L in 2008 to 5.3 mIU/L in 2018. During the 10-year review period, the proportion of patients being prescribed THRT for overt hypothyroidism increased significantly from 7.6% to 8.4% [26]. Despite this increase, at the endpoint of this study, more than 90% of patients that started THRT had only mild or no alterations of TFT at the time of THRT initiation. Our findings suggest a similar trend of a decreasing threshold for starting THRT for individuals with BD/SZD, despite limited evidence for any benefit observed. We did not see any difference between psychiatrists and GPs in the prescribing pattern.

Regarding hypothesis three, we had postulated that TSH_THRT_ would be lower in patients treated with lithium. As lithium-associated hypothyroidism is a well-known adverse effect, we had assumed that the clinicians’ threshold to start THRT would be lower [4,54,55]. In patients not treated with lithium, we had assumed clinicians would apply watchful waiting before starting THRT. However, we found that the median TSH_THRT_ was significantly lower in patients treated with a MS other than lithium. Our findings did not confirm hypothesis three.

Other MSs, particularly quetiapine, can also cause hypothyroidism [4], but they may not be monitored to the same extent as lithium. The prevalence of thyroid dysfunction will partly depend on the number of tests performed and possibly also on the testing interval. Hence, the risk of lithium-associated hypothyroidism relative to other MSs may be overestimated due to observation bias [4].

The effect of MS combination therapy on TSH concentrations remains largely unexplored.

Combination treatment is common within bipolar patients; 75–85% receive more than one MS [56,57]. In our study, we found a point prevalence of 53.8%. Hypothetically, combination treatment with different MSs could have an additive effect on thyroid function, leading to higher TSH concentrations. A study from Taiwan reported a 1.34 times increased risk of hypothyroidism for each additional MS prescribed [58]. A combination of MS and other psychotropic drugs, such as antidepressants or antipsychotics, may also alter TFT [59,60]. In our study, there was no difference in median TFT at THRT start between patients receiving one or several MSs. Neither had patients undergoing frequent treatment changes with lower median TSH_THRT_ than patients with stable treatments.

Our study has several strengths and limitations. Using routine clinical data, this observational study followed a large sample of patients with BD/SZD over a 21-year review period. The long follow-up time allowed trend analysis of TSH concentration at THRT start. The LiSIE cohort, on which this study is based, covered 84% of the eligible patients. Participating and non-participating patients were similar in terms of key parameters. We also manually validated BD/SZD diagnosis and all prescribing data from the medical records for all patients. This way, we could establish the chronology of adverse events with certainty, thereby minimising the risk of association bias.

The study was observational, relying on retrospectively collated information from medical records. Hence, the quality of our results depended on the quality of the documented clinical information. However, access to data at the symptom level reduced the potential for misclassification beyond what is possible in observational studies based on register data. To minimise misclassification and observer bias, we ensured that we (a) abstracted all data along with predefined variables, (b) counted only events that had been explicitly recorded in the medical notes, and (c) discussed all unclear events in the research group for consensus. At the same time, our study reflected a real-life clinical setting, generalizable in the absence of selection bias. We took great care to establish the chronology of events. However, since we relied on medical records from 1997, we may have missed earlier THRT initiations. Our trend analysis covering the period 1997–2017 might have been biased towards patients with longer times between MS and THRT initiation. Such a bias could have arisen as no patients were recruited into the study after 2012. Therefore, we performed a trend analysis of median TSH_THRT_ between 1997 and 2012. This trend analysis did not find a significant decrease in median TSH_THRT_. This could be due to the time reduction of 24% and the relatively modest annual decrease in TSH of 0.10 mIU/L. Finally, in our last hypothesis, we only included patients for whom we could reasonably assume that THRT had indeed been attributable to MS. We stratified MS treatment nearest THRT start with methods used by other studies [37]. However, THRT and MS starting dates were not always clearly recorded. In such cases, we had to extrapolate the likely starting dates from the medical records.

Prescribing THRT for patients with BD/SZD who only have mild or no alterations of TFT, has clinical implications. The evidence for this practice is limited. Findings from the general population suggest that prescribing THRT for subclinical hypothyroidism may offer no benefits. Subclinical hypothyroidism may reverse to normal, and THRT might not necessarily improve mood. However, this evidence may not apply to patients with BD/SZD. We do not know for certain how far medication-associated minor changes of thyroid function affects mental state in patients with a pre-existing affective disorder [38]. Once THRT is started, its prescription tends to be long-term. A UK retrospective cohort study based on the UK General Practice Databank followed THRT prescriptions from 2001 to 2009. Of those still in the study at the end of follow-up, 90% had received a THRT prescription in the last year [24]. Hence, it is important to consider the potential adverse effects of THRT such as osteoporotic fractures [61,62,63] and atrial fibrillation [63,64]. An indirect adverse effect from THRT could arise if THRT delayed or distracted from improving the treatment of the underlying affective disorder with MSs.

Clinical guidelines for patients with BD acknowledge that thyroid dysfunction can occur, particularly in the context of lithium treatment. However, guidelines do not give definite recommendations on how to treat subclinical hypothyroidism. Instead, most guidelines suggest that levothyroxine could be used and/or an expert opinion should be sought [65,66,67,68,69,70,71]. The UK National Institute for Health Care Excellence suggests considering levothyroxine for adults with subclinical hypothyroidism who have a TSH ≥ 10 mIU/L, on two separate occasions three months apart [69]. The Canadian Network for Mood and Anxiety Treatments propose levothyroxine or liothyronine as a third-line treatment for acute management of BD-II depression [68].

## 5. Conclusions

Our study shows that THRT may commonly be prescribed in patients with BD or SZD despite only mild or no TFT alterations and/or non-specific clinical symptoms attributable to hypothyroidism. As in the general population, the TSH threshold at which THRT is prescribed has become lower over time, this is also the case for patients with BD or SZD. The findings suggest that the reasons for prescribing THRT in this group of patients are heterogeneous and may partly reflect the complexity of the underlying mood disorder. THRT is rarely stopped once initiated. Hence it is important to monitor the clinical effect of THRT to minimise adverse effects that could occur over time. At the same time, it is important to be aware of a potential association bias that could arise from symptoms of BD and SZD being erroneously ascribed to mild changes in thyroid function. If THRT is mainly prescribed for psychiatric symptoms, mental status should be systematically assessed before and during THRT. This way, clinicians can ensure that THRT is not continued long-term in the absence of any benefits. In the future, guidelines need to address when and how to use THRT in these complex clinical scenarios.

## Figures and Tables

**Figure 1 jcm-10-05062-f001:**
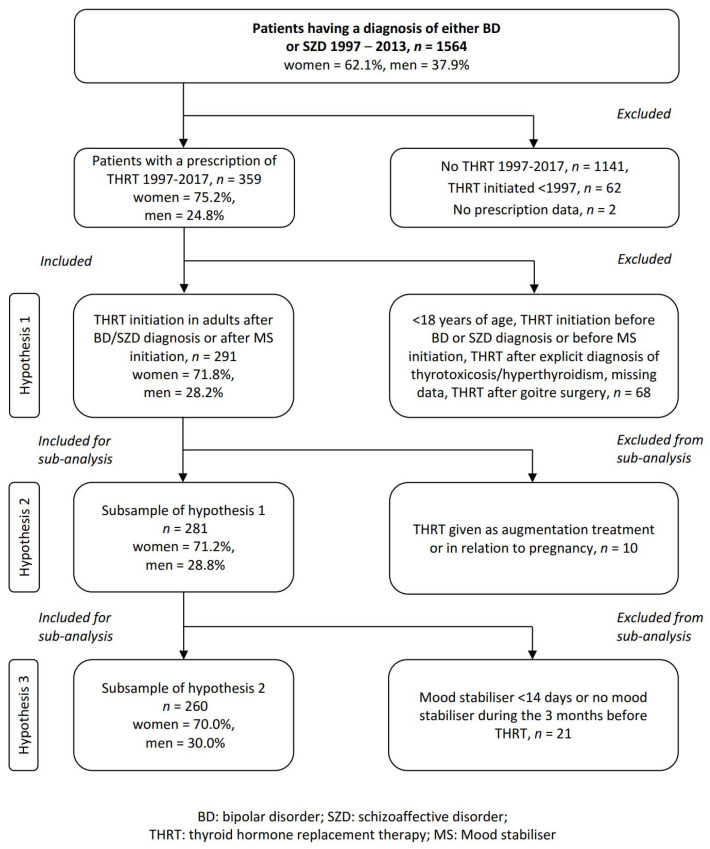
Selection of study sample.

**Figure 2 jcm-10-05062-f002:**
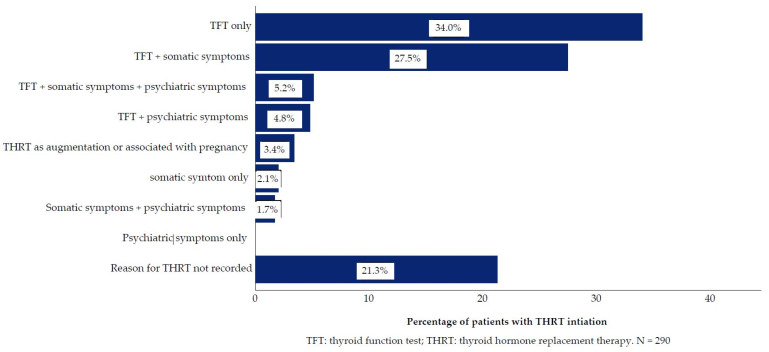
Decision drivers for Thyroid Hormone Replacement Therapy initiation.

**Figure 3 jcm-10-05062-f003:**
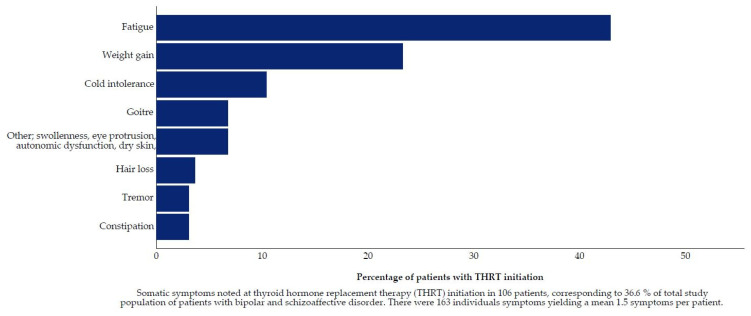
Somatic symptoms at Thyroid Hormone Replacement Therapy initiation.

**Figure 4 jcm-10-05062-f004:**
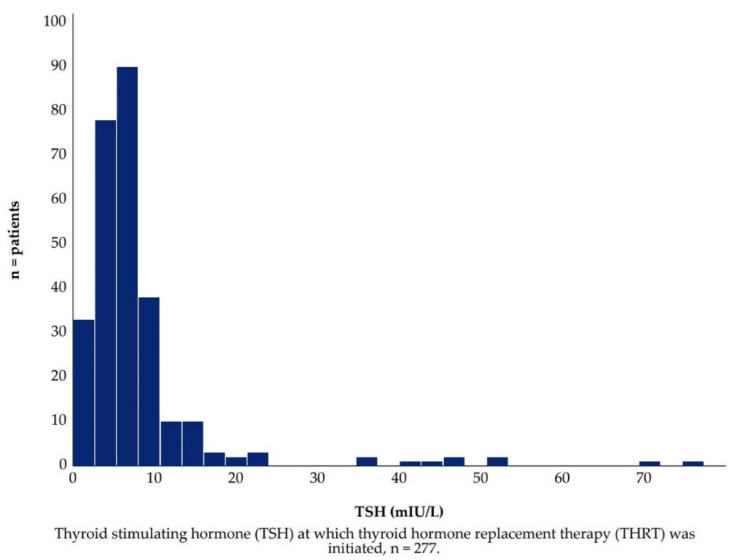
TSH distribution at Thyroid Hormone Replacement Therapy start.

**Figure 5 jcm-10-05062-f005:**
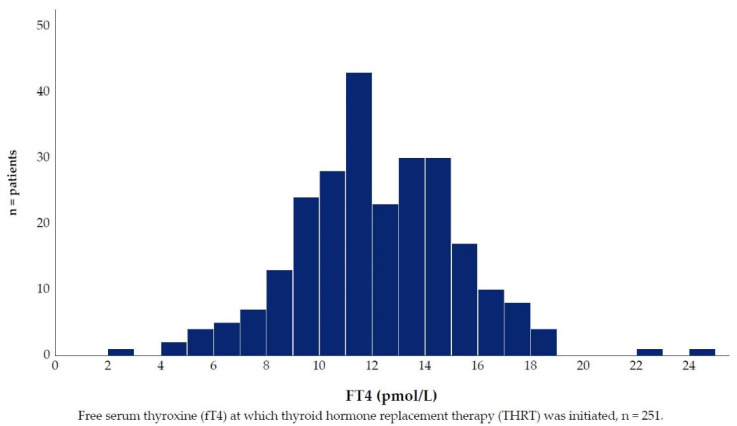
FT4 distribution at Thyroid Hormone Replacement Therapy start.

**Figure 6 jcm-10-05062-f006:**
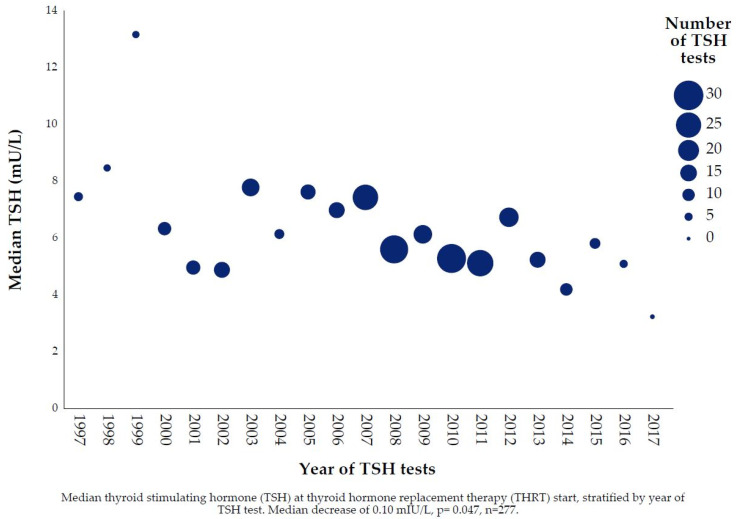
Time trend of median TSH at start of Thyroid Hormone Replacement Therapy.

**Figure 7 jcm-10-05062-f007:**
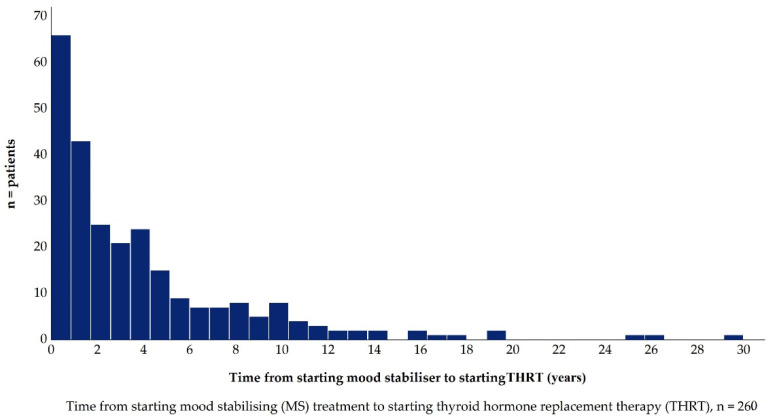
Time from starting MS to Thyroid Hormone Replacement Therapy.

**Figure 8 jcm-10-05062-f008:**
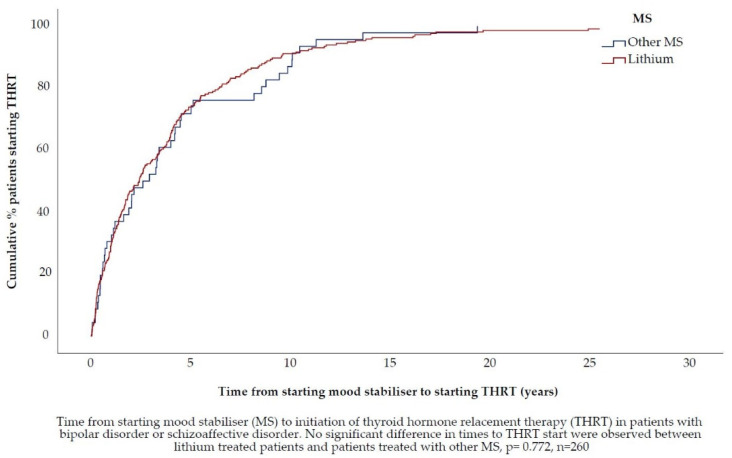
Time from starting Mood Stabiliser to initiating Thyroid Hormone Replacement Therapy, stratified by Mood Stabiliser.

**Table 1 jcm-10-05062-t001:** Categorisation of thyroid status at start of thyroid hormone replacement therapy.

Thyroid Status	Laboratory Values
Normal	TSH and fT4 within the normal reference interval
Hypothyroidism	
*Overt*	TSH elevated, fT4 lowered
*Subclinical*	
*Grade 1*	TSH elevated, <10, fT4 normal
*Grade 2*	TSH elevated, ≥10, fT4 normal
Low fT4	TSH normal, fT4 lowered
Unclassified/other	TSH with unknown fT4, fT4 with unknown TSH, lowered TSH with normal fT4, or elevated fT4 with normal TSH but no formal diagnosis of hyperthyroidism/thyrotoxicosis.

TSH: thyroid stimulating hormone; fT4: free serum thyroxine.

**Table 2 jcm-10-05062-t002:** Baseline characteristics.

	Hypothesis 1	Hypothesis 2	Hypothesis 3	
	*n* = 291	*n* = 281	*n* = 260	
			Patients with lithium *n* = 214	Patients with MS other than lithium *n* = 46
Sex, *n* (%)				
Women	209 (71.8) *	200 (71.2)	148 (69.2)	34 (73.9)
Men	82 (28.2) *	81 (28.8)	66 (30.8)	12 (26.1)
Age (years) at THRT start,				
median (min, max, IQR)	46.2 (19.5, 87.2, 22.2)	46.8 (19.5, 87.2, 23.9)	45.8 (19.5, 87.2, 19.5)	45.7 (19.8, 70.1, 18.9)
Age at THRT start, *n* (%)				
<60 years	245 (84.2)	235 (83.6)	177 (82.7)	42 (91.3)
≥60 years	46 (15.8)	46 (16.4)	37 (17.3)	4 (8.7)
Type of diagnosis, *n* (%)				
BD-I/SZD	101 (34.7)	100 (35.6)	78 (36.4)	15 (32.6)
BD-II/BD other	190 (65.3)	181 (64.4)	136 (63.6)	31 (67.4)
TSH_THRT_ available, *n* (%)	282 (96.9)	277 (98.6)	213 (99.5)	45 (97.8)
fT4_THRT_ available, *n* (%)	256 (88.0)	251 (89.3)	190 (88.8)	43 (93.5)
MS at time of				
THRT initiation, *n* (%)				
Lithium			214 (100.0)	0 (0.0) ***
Carbamazepine			5 (2.3)	3 (6.5)
Lamotrigine			30 (14.0)	15 (32.6) *
Valproate			36 (16.8)	17 (37.0) **
Aripiprazole			3 (1.4)	3 (6.5)
Olanzapine			41 (19.2)	10 (21.7)
Quetiapine			29 (13.6)	14 (30.4) *
Risperidone			14 (6.5)	6 (13.0)
MS combination at time of THRT initiation, *n* (%)				
Monotherapy			94 (43.9)	26 (56.5)
Combination therapy			120 (56.1)	20 (43.5)
Stability of MS treatment				
medication, *n* (%)				
Stable (≤1 changes)			151 (70.6)	34 (73.9)
Unstable (≥2 changes)			63 (29.4)	12 (26.1)
Time from MS start to THRT start				
Median (min, max, IQR)			2.4 (0.04, 29.2, 4.2)	2.8 (0.06, 19.7, 3.2)
Clinic initiating THRT, *n* (%)				
GP	107 (36.8)	104 (37.0)	60 (28.0)	29 (63.1)
Psychiatric clinic	170 (58.4)	164 (58.4)	144 (67.3)	15 (32.6)
Other	14 (4.8)	13 (4.6)	10 (4.7)	2 (4.3)
Other psychotropic				
drugs associated with				
hypothyroidism, *n* (%)				
Phenothiazines (derivates)	16 (5.5)	14 (5.0)	9 (4.2)	2 (4.3)
TCA	9 (3.1)	9 (3.2)	6 (2.8)	3 (6.5)

* *p* ≤ 0.01, ** *p* ≤ 0.001, *** *p* ≤ 0.0001. THRT: thyroid hormone replacement therapy; IQR: interquartile range; BD: bipolar disorder; SZD: schizoaffective disorder; TSH_THRT_: TSH at THRT initiation; fT4_THRT_: free serum thyroxine at THRT initiation; MS: mood stabiliser; GP: general practitioner; TCA: tricyclic antidepressants.

**Table 3 jcm-10-05062-t003:** Thyroid function at Thyroid Hormone Replacement Therapy initiation, based on thyroid function tests for the whole sample and according to type of bipolar disorder.

Thyroid Function, *n* (%)	Whole Sample	BD-I/SZD	BD-II/BD Other	*p*-Value
	291 (100)	101 (34.7)	190 (65.3)	
Overt hypothyroidism	76 (26.1)	26 (25.7)	50 (26.3)	0.916
Subclinical hypothyroidism, total	118 (40.5)	39 (38.6)	79 (41.6)	0.564
*Grade 1*	99 (34.0)	29 (28.7)	70 (36.8)	0.139
*Grade 2*	19 (6.5)	10 (9.9)	9 (4.7)	0.090
Low fT4	18 (6.2)	3 (3.0)	15 (7.9)	0.126
Normal	37 (12.7)	13 (12.9)	24 (12.6)	0.945
THRT as augmentation	7 (2.4)	1 (1.0)	6 (3.2)	0.428
Unclassified/other	35 (12.0)	19 (18.8)	16 (8.4)	0.014

THRT: thyroid hormone replacement therapy; BD: bipolar disorder; fT4: free serum thyroxine; SZD: schizoaffective disorder.

## Data Availability

The datasets generated and/or analysed during the current study are not publicly available due to lack of ethics committee permission and not having been part of the consent process. The structure of the dataset and the coding specification are available from the authors. Any other reasonable request will be raised with the Swedish Ethical Review Authority and the healthcare provider.

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
