# Peer review of "Patterns of Thyroid Hormone Prescription in Patients with Bipolar or Schizoaffective Disorder: Findings from the LiSIE Retrospective Cohort Study"

_jcm, 2021, doi:10.3390/jcm10215062_

Round 1

Reviewer 1 Report

As a clinically oriented psychiatrist, I very appreciate the topic. I am persuaded , that it is necessary to verify  some approaches which are broadly used however, are not evidence -based Especially, when taking in consideration, that in lack of new drugs with new target, we have to optimize the current available  treatment. I have nothing to add to this very useful a preciously done paper.

Author Response

Thank you kindly for your feedback.

Reviewer 2 Report

Thank you for the opportunity to review this paper. I have no comment regarding the content of this study, only formal clarification. I would like to commend the authors for their hard work in categorizing this data and performing the statistical data.

In the introduction section (page 1, lines 6-9 of paragraph 1) we discuss the "plausible" hypothesis of a link between hypothyroidism and depression. I think this deserves more detail, as explained a few pages later in the discussion section (page 14, lines 1-14), where conflicting data in the literature are discussed with links to references that are more relevant and inclusive. Finally, possible psychiatric symptoms of hypothyroidism are reported (page 4, lines 10-12 of paragraph 2.5.2), without specifying from where they have been taken or derived (missing references?) or if they have been plausibly hypothesized by the authors themselves.

On page 2, line 2 of the Aims paragraph, the acronym SZD is used, referring to Schizoaffective Disorder, but without it ever being introduced (excluding the abstract).

In the section "2.2 LiSIE participants", line 2, the diagnoses are formulated according to the ICD-10 nosographic manual (I suppose, because the number has been omitted in the paper), which does not make a clear distinction between Bipolar Disorder I or II. This distinction instead has been made by the authors on page 5, paragraph 2.6.1. It should be specified which nosography was used in this case.

References [22] and [23], page 2, section 2.2, may be irrelevant included in this section. It would be better to discuss them in another section, or specify if the sample used was also used in the cited studies.

It should also be specified, for scientific curiosity or reproducibility of the study, until when the collection of data from patients included in the study has been carried out, or if this is still in progress, despite having closed the recruitment of new subjects (last line of paragraph 2.2).

In the inclusion or exclusion criteria, page 3, there is no specification regarding Mood Stabiliser treatment, written instead in Figure 1: "</=14 days or no mood stabiliser during the 3 months before THRT". This detail is however described in section 2.6.4.

In Figure 1, section Hypothesis 2, on page 3, 281 subjects are included. However, it is described later (section 3.3.) how instead there are 277 subjects due to lack of data.

For further information, some references regarding TSH cut-off points would be needed (page 13, Paragraph 4, lines 7-9).

Author Response

Thank you kindly for your comments, please see the attachment for our response.
